# *Fallopia japonica* Root Extract Ameliorates Ovalbumin-Induced Airway Inflammation in a CARAS Mouse Model by Modulating the IL-33/TSLP/NF-κB Signaling Pathway

**DOI:** 10.3390/ijms241512514

**Published:** 2023-08-07

**Authors:** Juan Jin, Yan Jing Fan, Thi Van Nguyen, Zhen Nan Yu, Chang Ho Song, So-Yong Lee, Hee Soon Shin, Ok Hee Chai

**Affiliations:** 1Department of Anatomy, Jeonbuk National University Medical School, Jeonju 54896, Republic of Korea; jinjuan0619@gmail.com (J.J.); vandkh1993@gmail.com (T.V.N.);; 2Institute for Medical Sciences, Jeonbuk National University Medical School, Jeonju 54896, Republic of Korea; 3Department of Food Biotechnology, Korea University of Science and Technology, Daejeon 34113, Republic of Koreahsshin@kfri.re.kr (H.S.S.); 4Department of Food Functionality Research, Korea Food Research Institute, Wanju 55365, Republic of Korea

**Keywords:** combined allergic rhinitis and asthma syndrome, *Fallopia japonica* root extract, ovalbumin, mast cells, Th1 cytokines, Th2 cytokines

## Abstract

*Fallopia japonica* (Asian knotweed) is a medicinal herb traditionally used to treat inflammation, among other conditions. However, the effects of *F. japonica* root extract (FJE) on airway inflammation associated with combined allergic rhinitis and asthma (CARAS) and the related mechanisms have not been investigated. This study examined the effect of FJE against CARAS in an ovalbumin (OVA)-induced CARAS mouse model. Six-week-old male BALB/c mice were randomly segregated into six groups. Mice were sensitized intraperitoneally with OVA on days 1, 8, and 15, and administered saline, Dexamethasone (1.5 mg/kg), or FJE (50, 100, or 200 mg/kg) once a day for 16 days. Nasal symptoms, inflammatory cells, OVA-specific immunoglobulins, cytokine production, mast cell activation, and nasal histopathology were assessed. Administration of FJE down-regulated OVA-specific IgE and up-regulated OVA-specific IgG2a in serum. FJE reduced the production of T helper (Th) type 2 cytokines, and the Th1 cytokine levels were enhanced in nasal and bronchoalveolar lavage fluid. Moreover, FJE positively regulated allergic responses by reducing the accumulation of inflammatory cells, improving nasal and lung histopathological characteristics, and inhibiting inflammation-associated cytokines. FJE positively modulated the IL-33/TSLP/NF-B signaling pathway, which is involved in regulating inflammatory cells, immunoglobulin levels, and pro-inflammatory cytokines at the molecular level.

## 1. Introduction

Allergic asthma is a chronic and common respiratory disease that is a major health concern globally, affecting individuals of all ages, and is estimated to impact 339 million people worldwide. Another 100 million people are anticipated to be diagnosed with asthma by 2025 [1]. Allergic rhinitis (AR) is an allergic-mediated inflammatory condition that causes nasal irritation, congestion, and sneezing and is a key risk factor for asthma [2]. If AR is managed poorly, the risk of asthma increases several fold [3]. Recently, combined allergic rhinitis and asthma syndrome (CARAS) has emerged as a novel disorder associated with lower and upper lung inflammation [4]. AR and asthma are closely associated, as they are triggered by similar etiological factors, exhibit similar symptoms, and respond to equivalent therapeutic interventions [4]. Understanding the pathogenesis of CARAS is crucial for developing potential treatments. The pathogenesis of CARAS involves complex interactions between inflammatory eosinophils, T helper (Th) effector cells, IgE-activated mast cells, and free inflammatory cytokines.

Anti-histamines, anti-leukotrienes, decongestants, and nasal corticosteroids are widely used to treat AR [5]. However, they are only partially effective at suppressing AR-related symptoms and are frequently associated with side effects such as throat irritation, nasal dryness, dry mouth, headache, and dry eyes [2,6,7]. Thus, a safer and more effective compound for treating AR is desirable in patients with multiple disorders. Recently, there has been increasing demand for natural compounds that can be potentially applied as therapies against airway inflammation. Previous investigations have demonstrated the efficacy of polyphenols against inflammation and oxidation, which are critical to the development of respiratory disorders [8,9,10]. Polyphenols suppress allergen-induced inflammatory cell infiltration, serum IgE level, and inflammatory cytokines interleukin (IL)-4, IL-5, and IL-13 in bronchoalveolar lavage fluid (BALF) and inhibit histamine release from mast cells to induce anti-inflammatory effects in patients with airway disorders [11,12,13]. Identifying polyphenol-rich natural materials is necessary for developing novel, safe, and effective compounds against AR. Supplementation with natural polyphenols may potentially assist in preventing airway hyper-responsiveness.

*Fallopia japonica* (Asian knotweed) is a traditional medicinal herb native to eastern China, Korea, Japan, Taiwan, and eastern Russia. Traditionally, *F. japonica* has been used to treat jaundice, cough, inflammation, digestive problems, favus, scald, and allergic inflammatory diseases. In addition, it has been utilized to enhance blood circulation, treat bronchitis, and eliminate phlegm [14,15,16]. *F. japonica* is rich in polyphenols such as resveratrol, flavones/flavonol, polydatin, and glycosides [17,18,19]. *F. japonica* contains important anthraquinones such as emodin, fallacinol, and physcion, which suppress inflammation by inhibiting leukocyte movement and preventing β-cell destruction [20]. Extracts of *F. japonica* may inhibit the effects of tumor necrosis factor (TNF)-α, probably due to the presence of resveratrol in the extract [21]. Resveratrol significantly affects the modulation of inflammatory processes [18]. However, the influence of *F. japonica* against CARAS has not been investigated in depth. Moreover, *F. japonica* root extract (FJE) can renew the challenge of addressing the disorder. In this study, we evaluated the influence of FJE against CARAS. Our results reveal a possible mechanism underlying the positive influence of FJE on AR and asthma.

## 2. Results

### 2.1. FJE Suppressed Rat Peritoneal Mast Cell (RPMC) Degranulation

To examine the influence of FJE on mast cells, which play a critical role in anti-allergic inflammatory responses, RPMC degranulation was assessed. Compound 48/80 (C48/80) leads to mediator exocytosis and RPMC degranulation by stimulating the signal transduction pathway. We observed that normal RPMCs had packed secretory granules, were oval, and showed a clear cell outline. RPMCs treated with FJE revealed characteristics similar to normal RPMCs, irrespective of the concentration used. However, RPMCs induced with C48/80 showed indistinct cell outlines, and the expression of intracytoplasmic granules on the cell surface indicated substantial degranulation (Figure 1A). FJE supplementation at 0.1–10 mg/mL after stimulation with C48/80 suppressed the degranulation of mast cells (Figure 1B).

### 2.2. FJE Inhibited Nasal Symptoms in the OVA-Induced CARAS Mouse Model

The frequencies of nasal rubbing and sneezing were recorded to examine the effects of FJE on the OVA-induced CARAS mice. Mice in the CARAS group revealed significantly higher rates of nasal rubbing and sneezing than those in the control group. Notably, FJE reduced the sneezing and rubbing incidences in a dose-dependent manner, with significantly lower numbers of events than in the CARAS group (Figure 1C,D).

### 2.3. FJE Reduced Infiltration of Inflammatory Cells in the Nasal Lavage Fluid (NLF) and BALF of the OVA-Induced CARAS Mouse Model

Activation of the inflammatory response plays an essential role in the development of CARAS. To assess the anti-inflammatory effects of FJE on CARAS, inflammatory cells were quantified in NLF and BALF. As shown in Figure 2A,D, the total number of inflammatory cells was significantly higher in the CARAS group than in other groups. Administration of FJE led to a dose-dependent reduction in inflammatory cells; groups that received FJE doses of 100 and 200 mg/kg had a significantly lower number of inflammatory cells than the CARAS group. Moreover, the CARAS group had greater numbers of epithelial cells, macrophages, eosinophils, neutrophils, and lymphocyte cells than other groups. The FJE-treated group demonstrated a dose-dependent reduction in the numbers of epithelial cells, macrophages, eosinophils, neutrophils, and lymphocytes (Figure 2B,C,E,F). Collectively, these observations suggest that FJE suppressed the total number of inflammatory cells in the NLF and BALF. These observations indicate the positive influence of FJE on the airway inflammation of the CARAS mice.

### 2.4. FJE Suppressed Histopathological Alterations in the Nasal Mucosa and Lung Tissue of the OVA-Induced CARAS Mouse Model

Nasal mucosal thickness was significantly greater in the CARAS group than in the other groups. Mice administered with FJE and Dexamethasone (Dex) showed less mucosal edema than CARAS mice. Moreover, the subepithelial layer of the nasal mucosa was significantly thinner in mice treated with 200 mg/kg of FJE and Dex than in the CARAS mice (Figure 3A–C). Furthermore, the bronchus showed significant changes in the mice of the CARAS group compared with control mice (Figure 3E). PAS staining showed less goblet cell hyperplasia in FJE- and Dex-treated mice than in the CARAS mice. Interestingly, OVA-activated goblet cells exhibited mucus hypersecretion in the nasal and lung tissues of CARAS mice. Dex and FJE (100 and 200 mg/kg) supplementation suppressed mucus overproduction (Figure 3F). Additionally, toluidine blue staining of mast cells showed higher numbers in CARAS mice compared with FJE-treated mice (Figure 3G), whereas, in the CARAS mice, there was a greater number of eosinophils. Eosinophils are stained red in the cytoplasm and marked with black head arrows in Figure 3H. These findings strongly indicate that FJE administration alleviated pathological injury in mice with OVA-induced CARAS.

### 2.5. FJE Suppressed Immunoglobulin Levels in Serum of the OVA-Induced CARAS Mouse Model

To investigate whether FJE affects allergic reactions, OVA-specific IgE, IgG1, and IgG2a in serum were measured in the mice with OVA-induced CARAS. OVA-specific IgE and OVA-specific IgG1 levels were notably elevated in the CARAS mice compared to the control group. However, FJE and Dex significantly controlled this upregulation (Figure 4A,B). OVA-specific IgG2a in FJE 200 and Dex groups was significantly improved compared with the CARAS group (Figure 4C).

### 2.6. FJE Regulated T Helper–Associated Cytokine Production in NLF and BALF of the OVA-Induced CARAS Mouse Model

The levels of Th2-associated cytokines IL-4, IL-5, IL-6, and IL-13 in both NLF and BALF were significantly up-regulated in the CARAS group compared with the control group. FJE 200 mg/kg and Dex were more efficient in suppressing these Th2-associated cytokines than the other treatments. However, the levels of IL-5 in the NLF were insignificant in all groups except FJE 200 mg/kg (Figure 5D). Similarly, the levels of IL-4, IL-5, IL-6, and IL-13 were significantly higher in the BALF of the CARAS group, whereas FJE and Dex suppressed the levels of Th2-associated cytokines (Figure 5). The levels of Th1-associated cytokines IL-12 and IFN-γ levels in both NLF and BALF were significantly lower in the CARAS group than in the control group. However, treatments with FJE and Dex improved IFN-γ and IL-12 levels in the NLF and BALF (Figure 5A,B,H,I). Moreover, TNF-α, a pro-inflammatory cytokine, was also assessed to determine the effect of FJE on CARAS. The TNF-α levels in both NLF and BALF were significantly higher in the CARAS group than in the other treated groups. FJE dose-dependently down-regulated TNF-α, whereas levels were significantly lower in the FJE-treated groups than in the CARAS group.

### 2.7. FJE Regulates Epithelium-Derived Cytokines in NLF and BALF of the OVA-Induced CARAS Mouse Model

Epithelium-derived cytokines such as thymic stromal lymphopoietin (TSLP) and IL-33 play vital roles in linking innate and adaptive immune reactions associated with Th2 cytokine-mediated reactions [22,23]. TSLP and IL-33 are also potentially involved in the pathogenesis of asthma [23]. Hence, TSLP and IL-33 levels were measured to assess the effect of FJE on CARAS. TSLP was significantly higher in the CARAS group; however, FJE treatment significantly reduced these levels in both NLF and BALF. Notably, the administration of various concentrations of FJE resulted in similar effects on the TSLP level. This suggests that the lowest concentration was adequate in regulating the TSLP level (Figure 6A,C). However, in BALF, a dose-dependent regulation was observed. Furthermore, the level of IL-33 was significantly elevated in the NLF and BALF of the CARAS group. However, FJE and Dex significantly reduced these levels, to a greater degree than that in the CARAS group (Figure 6B,D). Notably, the IL-33 level in the lung tissue clearly indicates the dose-dependent regulatory effects of FJE (Figure 6E). Moreover, immunohistochemistry revealed significantly lower expression of ST2, the receptor for IL-33, in nasal tissues upon FJE and Dex treatment compared with levels in the CARAS group (Figure 6F).

### 2.8. FJE Regulates the Inflammatory Response via the NF-κB Signaling Pathway in the OVA-Induced CARAS Mouse Model

Activation of NF-κB signaling is critical to the regulation of Th2 cytokine levels [24]. To determine the effect of FJE on mice with OVA-induced CARAS, NF-κB signaling was evaluated. The levels of P-NF-κB and P-IκB in NLF were significantly higher in the CARAS group than in the control group, while FJE down-regulated levels. Similar outcomes were observed in BALF (Figure 7A–F). The NF-κB-related proteins, such as P-NF-κB and P-IκB, were noticeably expressed in the lung tissues of CARAS mice compared with control mice; in contrast, these were significantly reduced in FJE and Dex-treated mice (Figure 7G).

## 3. Discussion

CARAS is a novel disorder linked to lower and upper lung inflammation. This combined inflammation immediately induces a coordinated response by mast cells and their mediators, influencing nasal congestion and vascular permeability. The use of polyphenol-rich natural sources could reduce the potential repercussions associated with the coordinated response. Therefore, in this study, we examined the effect of polyphenol-rich FJE on CARAS and the associated mechanism in a mouse model. FJE inhibited symptoms, such as rubbing and sneezing, consistent with prior results in placebo-controlled trials showing that polyphenols relieved nose symptoms [25,26]. Further, the release of cytokines or inflammatory mediators causes the infiltration of inflammatory cells into the nasal mucosa, influencing the inflammatory response [27]. Airway inflammation is a crucial factor for the development of asthma and AR, as it causes morphological alterations in the airway via the infiltration of inflammatory cells such as neutrophils, eosinophils, and macrophages [28]. In this study, histopathological analysis revealed reduced mucosal edema, goblet cell hyperplasia, and a thinner subepithelial layer in the CARAS mouse model treated with FJE, suggesting reduced pathological injury. Consistent with previous reports [29], these findings demonstrate the efficacy of FJE in reducing OVA-induced inflammatory responses. Based on these results, we suggest that FJE exerts a protective effect on the OVA-induced CARAS mouse model.

Allergic mouse models have established a significant association between antigen-specific IgE and IgG1 [30]. These immunoglobulins are markers for Th2 cells, whereas IgG2a is a marker for Th1 cells [31]. In this study, FJE suppressed OVA-specific IgE and IgG1 while stimulating IgG2a. Therefore, a higher IgG2a/IgG1 ratio is expected with FJE treatment, as a higher IgG2a/IgG1 ratio indicates a protective immune response in allergic inflammation diseases [32]. Furthermore, mast cell degranulation releases inflammatory mediators such as histamines, prostaglandins D2, kinins, heparin, and serine proteases [33]. In this study, the administration of FJE significantly inhibited mast cell degranulation, which may have contributed to decreased rubbing and sneezing in FJE-treated mice. In contrast, the control group exhibited the maximum degranulation rate, with increased rubbing and sneezing activities, because enhanced vascular permeability and localized edema block nasal airways and caused congestion [34]. These observations suggest that FJE exhibited an anti-allergic effect in the OVA-induced CARAS model.

Research into mitigating airway inflammation led to the discovery of critical roles for IL-33, Th2, Th1 cytokines, TNF-α, NF-κB, and TSLP in the development of allergic airway diseases. The imbalance between Th1 and Th2 cell-mediated immunity plays a crucial role in the pathophysiology of AR. Increased Th2 cell-secreted cytokines trigger B cells to generate allergen-specific IgE, which causes acute symptoms. IL-4 stimulates IgE synthesis, IL-5 is involved in eosinophilic growth and differentiation, and IL-13 is involved in mucus production and airway hyper-reactivity [35,36]. In this study, FJE administration reduced IL-4, IL-5, and IL-13 levels, which influenced IgE synthesis, mucus production, and eosinophilic growth and differentiation. Furthermore, Treg cells potentially promote and maintain allergy tolerance by modulating both the innate and adaptive immune responses triggered by allergens [37]. However, FOXP3, a transcriptional regulator of Treg cells, is key to suppressing Th2 responses upon exposure to allergens [38]. Treg cells exert an immunomodulatory effect by secreting anti-inflammatory cytokines such as IL-10, restricting the activity of different immune cells and ultimately reducing the immune responses. Here, FJF treatment significantly reduced Th2 cytokines such as IL-4, IL-5, and IL-13 and improved the IL-10 levels in BALF and lung homogenate (Appendix A), suggesting improved Treg FOXP3. Similarly, previous investigations demonstrated that herbal extracts suppressed Th2 cytokines [39]. These reports strongly support the FJE study observations. Moreover, the downregulation of these cytokines was observed in both NLF and BALF after FJE treatment, confirming the influence of FJE. Moreover, the levels of Th2 cytokines such as IL-4 and IL-5 are significantly higher in patients with AR [40], which potentially inhibits the functions of the Th1 immune response [41]. Th1 cytokines, such as IFN-γ and IL-12, are essential for the development of allergic airway inflammation. IFN-γ and IL-12 are considered anti-inflammatory as they prevent allergies and manifestations of allergic inflammation [42]. Following these observations, FJE treatment stimulated IFN-γ and IL-12 levels in the OVA-induced CARAS mouse model.

NF-κB is a transcription factor responsible for various immune and inflammatory responses [43]. The NF-κB signaling pathway plays a role in airway inflammation, and an elevated NF-κB level has been frequently observed in OVA-induced asthmatic mice. NF-κB contributes to cytokine production and plays a vital role in the development of inflammatory diseases [44,45,46]. Additionally, NF-κB is involved in T cell differentiation, Th0 differentiation, and IL-4 induction, affecting the differentiation of naive T cells and mediating the induction of pro-inflammatory cytokines such as IL-1β, IL-6, and TNF-α. These pro-inflammatory cytokines then activate NF-κB [47]. Thus, we examined NF-κB and P-NF-κB and P-IκB, components of the NF-κB signaling pathway, to explore their involvement in the FJE-induced protective effect. We found that FJE efficiently controlled the expression of P-NF-κB and P-IκB, suggesting an influence on cytokine production. Our observations suggest that FJE is approximately as effective as Dex at inhibiting airway inflammation.

TSLP, a pleiotropic cytokine, is a key factor in the pathogenesis of asthma, and elevated TSLP levels have been correlated with enhanced levels of TNF-α and Th2 cytokines [48]. TSLP is produced by epithelial cells and is triggered by various environmental stimuli; it plays a critical role in producing Th2 cytokines such as IL-33 and IL-25 and inducing Th2-type differentiation of CD4+ T cells. These associations influence the adaptive immune responses affecting respiratory disorders [24]. Additionally, TSLP is regulated by inflammatory mediators and TNF-α in an NF-κB-dependent manner [49]. In our study, reduced levels of TSLP were observed and correlated with lower levels of TNF-α and Th2 cytokines. These events may be involved in the underlying molecular mechanism of mitigating respiratory disorders by FJE. Additionally, ST2 is linked to asthma susceptibility, and recent investigations have shown that epithelial cells, endothelial cells, mast cells, and airway smooth muscle cells express the ST2 receptor [50,51]. IL-33 binding to ST2 recruits IRAK, IRAK4, MyD88, and TRAF6. ST2 activates NF-κB and MAP kinases, triggering pro-inflammatory effects [50,52]. Thus, inhibiting ST2-mediated inflammation may reduce the aggravation of airway inflammation. We found that administration of FJE significantly lowered ST2 expression. These observations confirm the potential effectiveness of FJE on CARAS. Moreover, all these positive implications of FJE can be attributed to the presence of EMODIN glucoside 1, EMODIN glucoside 2, procyanidin, and other polyphenols. The ultra-performance liquid chromatography data indicate that FJE contains a substantial quantity of these biologically active compounds (Appendix A). These compounds are known to suppress inflammation by inhibiting leukocyte movement, preventing β-cell destruction and alleviating airway inflammation [20,53,54,55]. In addition, procyanidins assist in prevention of airway inflammation [56,57,58]. Together, resveratrol’s potent anti-inflammatory and antioxidant properties and EMODIN’s anti-inflammatory properties help to reduce airway inflammation. In addition, Procyanidin’s potential to improve vascular health indirectly contributes to optimal oxygen transport and minimizing inflammation-associated damage [59,60,61]. Hence, the presence of these biologically active compounds positively influences multiple signaling pathways and regulates airway inflammation-associated disorders.

CARAS is a unified airway disease characterized by lower and upper lung inflammation with AR and asthma. The symptoms include airway hyper-responsiveness, mucus hypersecretion, and eosinophilic infiltration in the airways. Our data indicate that FJE exerts a protective effect against OVA-induced CARAS in mice by reducing the infiltration of inflammatory cells, regulating the accumulation of mucus, suppressing OVA-specific immunoglobulins, and modulating the IL-33/TSLP/NF-κB signaling pathway (Figure 8). The major observations of this study are consistent with those of pathways observed in human clinical trials. Future studies on cytokines such as IL-17 and the involvement of immune cells such as B cells, lymphoid, and NKT cells are required.

## 4. Materials and Methods

### 4.1. Preparation of Fallopia Japonica Root Extract

FJE was provided by the Natural F&P Co., Ltd. (Cheongju-si, Republic of Korea). Briefly, *F. japonica* plants were harvested from Jecheon-si (Chungcheong-do, Republic of Korea) and Youngcheon (Gangwon, Republic of Korea). Raw *F. japonica* was dried, pulverized, and extracted with 10 times the volume of distilled water by heating (±50 °C) for 6 h. FJE was filtered using a cartridge filter and concentrated using a Rotavapor R-210 (BÜCHI Labortechnik AG, Flawil, Switzerland) at 50 °C in a vacuum, followed by drying.

### 4.2. Animal Studies

Five week old male BALB/c mice were obtained from the Damool Science Co. (Daejeon, Korea). Animals were housed under pathogen-free conditions with a 12 h:12 h light/dark cycle with free access to food and water. The animal experiments were carried out following the Jeonbuk National University Animal Care and Use Committee guidelines (CBNU 2021-0115).

### 4.3. CARAS Model and Treatment

Animals were acclimatized for 1 week before use. Six-week-old mice were randomly segregated into 6 groups, with 6 mice in each group. Group 1 was the control group, which was sensitized, treated, and challenged with saline; Group 2 was the CARAS group, in which mice were sensitized and challenged with OVA; Groups 3–5 were the FJE groups, in which CARAS mice were treated with 50, 100, or 200 mg/kg of FJE; and Group 6 was the Dex group, in which CARAS mice received 1.5 mg/kg of Dex (D4902, Sigma, St. Louis, MO, USA). All test compounds were administered through oral gavage. The CARAS and treatment groups were sensitized on days 1, 8, and 15 with an intraperitoneal injection of 200 µL saline containing 50 μg OVA (#A5503, Sigma, St. Louis, MO, USA) and 1 mg aluminum hydroxide (#77161, Thermo Scientific, Rockford, MD, USA). On days 22–24, all mice received 5% OVA via ultrasonic nebulization. On days 25–31, mice received an intranasal challenge with 20 µL OVA solution (10 mg/mL); mice in the control group received saline instead of OVA. From day 16 to the end of the experiment (day 30), mice were administered FJE or Dex 1 h before intranasal challenge via oral gavage (day 31). On day 31, soon after the OVA challenge, rubbing and sneezing were recorded in each mouse for 15 min. The mice were sacrificed 24 h after the last OVA challenge. A schematic of the experimental overview is shown in Figure 9.

### 4.4. Preparation of RPMC and Degranulation Assay

RPMCs were isolated as described in previous reports [62]. Purified RPMCs were resuspended in HEPES-Tyrode buffer and incubated with 25 µL of saline or 25 µL of FJE (50, 100, 200 mg/kg) at 37 °C for 10 min. Cells were then incubated with compound 48/80 (C48/80, 5 μg/mL) or saline for 15 min. Finally, RPMCs were observed under a microscope (Nikon ECLIPSE Ti-U, Melville, NY, USA) and analyzed for mast cell degranulation.

### 4.5. Collection of Serum, NLF, and BALF

Mice were anaesthetized with diethyl ether 24 h after the last challenge, and blood samples were collected from eye sockets. The collected blood was centrifuged at 10,000 rpm/min at 4 °C to obtain serum. NLF was collected as described previously using an 18-gauge catheter. Briefly, the trachea was partly eviscerated; the catheter was inserted to perfuse phosphate-buffered saline (PBS), and the collected NLF was centrifuged. The supernatant was used for ELISA, the cell pellet was suspended in PBS, and total cell numbers were counted using a cytospin (TH-CYTO4, Thermo Electron Corporation, Waltham, MA, USA, Diff-Quik staining (38,721, Sysmex, Chuo-ku, Kobe, Hyogo, Japan), which was performed following the manufacturer’s instructions.

### 4.6. Histological Analysis

For histopathology examination, the head tissues were decalcified with a decalcifying solution (HS-105, National Diagnostics, Atlanta, GA, USA). Then the head and lung tissues were dissected and fixed in 10% neutral buffered formalin at 24 ± 2 °C for 3 days. To evaluate general morphology and goblet cell hyperplasia, and observe mast cell infiltration and eosinophils infiltration, paraffin-embedded tissues were cut into 5 µm sections and stained with hematoxylin and eosin (H&E) (#517-28-2, Sigma, St. Louis, MO, USA), periodic acid-Schiff (PAS) (#ab245886, Abcam, Cambridge, UK), toluidine blue (TB) (T3260, Sigma, St. Louis, MO, USA) and Sirius red (SR) (365548, Sigma, St. Louis, MO, USA). The lung inflammation score was measured as previously reported [63]. Goblet cells, mast cells, and eosinophils were measured using Fiji software.

### 4.7. Immunohistochemistry

Immunohistochemistry was performed using a Rabbit specific HRP/DAB (ABC) Kit (ab64261, Abcam, Cambridge, UK) and anti-ST2 antibody (MBS7604494, MyBioSource, San Diego, CA, USA). Briefly, the sections were brought to boiling temperature in 1× citrate buffer (ab93678, Abcam, Cambridge, UK), pH 6.0, and maintained at a sub-boiling temperature for 10 min for antigen retrieval. Sections were incubated with a blocking solution and incubated overnight at 4 °C with an anti-ST2 antibody (1:500). The sections were subsequently incubated with a biotinylated goat anti-rabbit IgG secondary antibody. Staining was developed using DAB solution for 1 min (1:50; Millipore, Billerica, MA, USA). The ST2-positive area was measured using the Fiji application.

### 4.8. Evaluation of Cytokine Levels and Immunoglobulins

Levels of IL-4 (#M4000B, R&D Systems, Inc., Minneapolis, MN, USA), IL-5 (#M5000, R&D Systems, Inc., Minneapolis, MN, USA), IL-6 (#M6000B, R&D Systems, Inc., Minneapolis, MN, USA), IL-12 (#M1270, R&D Systems, Inc., Minneapolis, MN, USA), IL-13 (#CSB-E04602m, CUSABIO, Houston, TX, USA), TSLP (#MTLP00, R&D Systems, Inc., Minneapolis, MN, USA), TNF-α (#MTA00B, R&D Systems, Inc., Minneapolis, MN, USA), IL-33 (#M3300, R&D Systems, Inc., Minneapolis, MN, USA), and interferon-gamma (#IFN-γ, MIF00, R&D Systems, Inc., Minneapolis, MN, USA) in NLF and BALF were assessed using ELISA kits. Levels of NF-κB p65, phospho-NF-ĸB p65 (#7174C, #7173CA, Cell Signaling, Danvers, MA, USA), and P-IκB (MBS2600002, MyBioSource, San Diego, CA, USA) in NLF and BALF were measured using commercial ELISA kits (#7174C, #7173CA, Cell Signaling, Danvers, MA, USA) as directed by the manufacturer. OVA-specific IgE (#439807, BioLegend, Inc., San Diego, CA, USA), anti-ovalbumin IgG1 (#500830, Cayman, Ann Arbor, MI, USA), and IgG2a (#3015, Chondrex, Woodinville, WA, USA) were quantified in serum using ELISA kits.

### 4.9. Western Blot Anlaysis

Collected lung tissues were homogenized and lysed with RIPA buffer. Protein samples (50 μg) were separated by 10% SDS-PAGE and transferred onto a PVDF membrane, which was then blocked with 5% skimmed milk at room temperature for 1 h and incubated with primary antibodies against NF-κB (1:1000; #8242, Cell Signaling Technology, Danvers, MA, USA), P-NF-κB (1:1000; #3031, Cell Signaling Technology, Danvers, MA, USA), IκB (1:1000, #9242, Cell Signaling Technology, Danvers, MA, USA), P-IκB (1:1000, #2859, Cell Signaling Technology, USA), IL-33 (1:1000, #88513, Cell Signaling Technology, Danvers, MA, USA), anti-rabbit IgG (1:1000, #7074, Cell Signaling Technology, USA), and β-actin (1:1000, #4970, Cell Signaling Technology, USA). The signals were detected and visualized using an enhanced chemiluminescence (ECL) detection system. Bands were analyzed using Image J software (NIH, Bethesda, Rockville, MD, USA).

### 4.10. Statistical Analysis

All the statistical analyses were performed with GraphPad Prism version 8.4 (GraphPad Software, Boston, MA, USA). Data are expressed as mean ± SEM. Student’s *t*-test and one-way ANOVA with Dunnett’s test were used to determine group differences. The significance level was set at *p* < 0.05.

## 5. Conclusions

Our findings provide substantial evidence for the anti-asthmatic and anti-inflammatory effects of FJE on an OVA-induced CARAS mouse model. FJE attenuated airway inflammation by suppressing inflammatory cells and decreasing serum anti-OVA IgE, anti-OVA IgG1 and pro-inflammatory cytokines in mice with OVA-induced CARAS. At the molecular level, FJE ameliorated CARAS by modulating the IL-33/TSLP/NF-κB signaling pathway. These findings suggest that FJE is a promising therapeutic functional food for airway disorders.

## Figures and Tables

**Figure 1 ijms-24-12514-f001:**
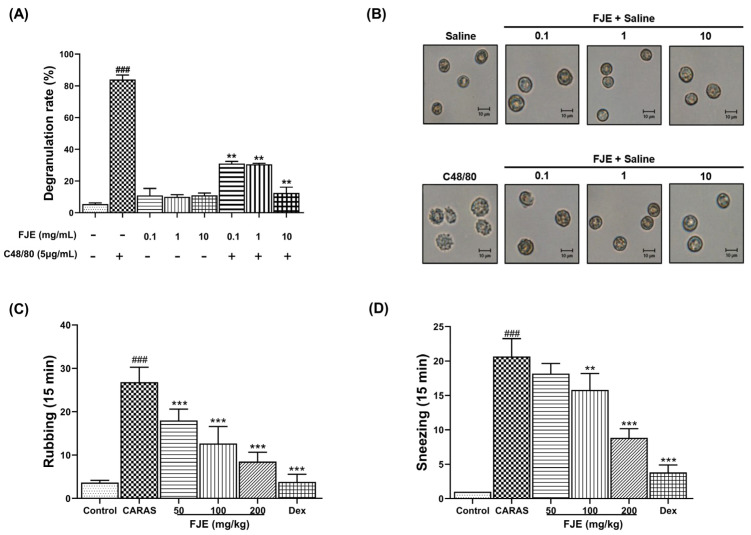
Influence of FJE on RPMC degranulation and nasal symptoms in mice with OVA-induced CARAS. (**A**) Degranulation rates upon co-treatment with C48/80 and FJE. (**B**) Microscopic observations in RPMCs treated with either C48/80 or FJE. (**C**) Nasal rubbing and (**D**) sneezing were counted for 15 min after the final OVA challenge on day 31. Mice were administered saline, Dex (1.5 mg/kg), or FJE (50, 100, or 200 mg/kg) daily for 16 days. Data represent mean ± SEM (*n* = 6/group). ### *p* < 0.001 compared with control, ** *p* < 0.01, *** *p* < 0.001 vs. CARAS. Dex, Dexamethasone.

**Figure 2 ijms-24-12514-f002:**
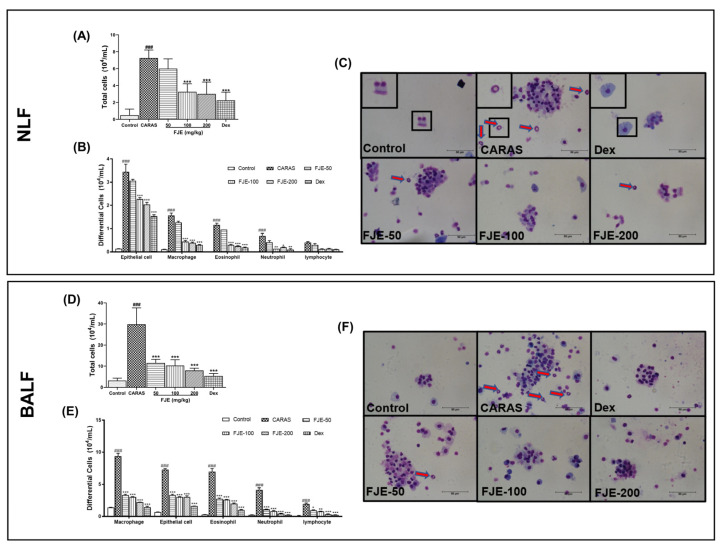
Infiltration of inflammatory cells in the NLF and BALF of mice with OVA-induced CARAS. (**A**) The total number of inflammatory cells. (**B**) Differential cell count (epithelial cells, macrophages, eosinophils, neutrophils, and lymphocytes) in NLF and (**C**) Diff-Quik-stained images. (**D**) The total number of inflammatory cells. (**E**) Differential cell count (epithelial cells, macrophages, eosinophils, neutrophils, and lymphocytes) in BALF and (**F**) Diff-Quik-stained images. Scale bar = 50 µm. Red arrows indicate eosinophils. Mice were administered saline, Dex (1.5 mg/kg), or FJE (50, 100, or 200 mg/kg) once a day for 16 days. Data represent mean ± SEM (*n* = 6/group). ### *p* < 0.001 compared with control, * *p* < 0.05, ** *p* < 0.01, *** *p* < 0.001 vs. CARAS. Dex, Dexamethasone.

**Figure 3 ijms-24-12514-f003:**
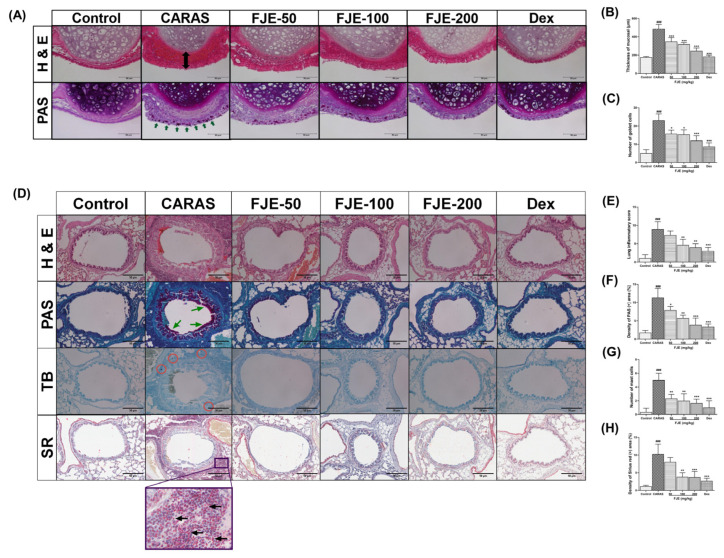
Histological analysis of nasal and lung tissues. (**A**) H&E- and PAS-stained nasal tissues. (green arrows indicate goblet cells) (**B**) Measurement of mucosal thickness (double-headed arrow) in H&E-stained nasal tissues. (**C**) Quantification of goblet cells in PAS-stained nasal tissues. (**D**) H&E-, PAS-, TB-, and SR-stained lung tissues. (green arrows indicate goblet cells) (**E**) Determination of lung inflammation by lung inflammation scoring using H&E-stained images. (**F**) Analysis of goblet cells using PAS-stained images. (**G**) Quantification of mast cells with TB-stained tissues (red circles indicate mast cells). (**H**) Analysis of eosinophils using SR-stained tissues (black arrows indicate eosinophils). Scale bar = 50 µm. Mice were administered saline, Dex (1.5 mg/kg), or FJE (50, 100, 200 mg/kg) daily for 16 days. Data represent mean ± SEM (*n* = 6/group). ### *p* < 0.001 compared with control, * *p* < 0.05, ** *p* < 0.01, *** *p* < 0.001 vs. CARAS. Dex, Dexamethasone.

**Figure 4 ijms-24-12514-f004:**
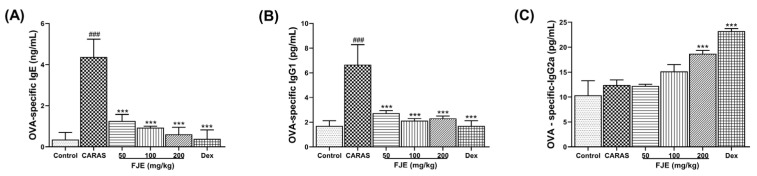
The expression of OVA-specific immunoglobulins in serum. Effects of FJE on levels of (**A**) OVA-specific IgE, (**B**) OVA-specific IgG1, and (**C**) OVA-specific IgG2a. Mice were administered saline, Dex (1.5 mg/kg), or FJE (50, 100, or 200 mg/kg) daily for 16 days. Data represent mean ± SEM (*n* = 6/group). ### *p* < 0.001 compared with control, *** *p* < 0.001 vs. CARAS. Dex, Dexamethasone.

**Figure 5 ijms-24-12514-f005:**
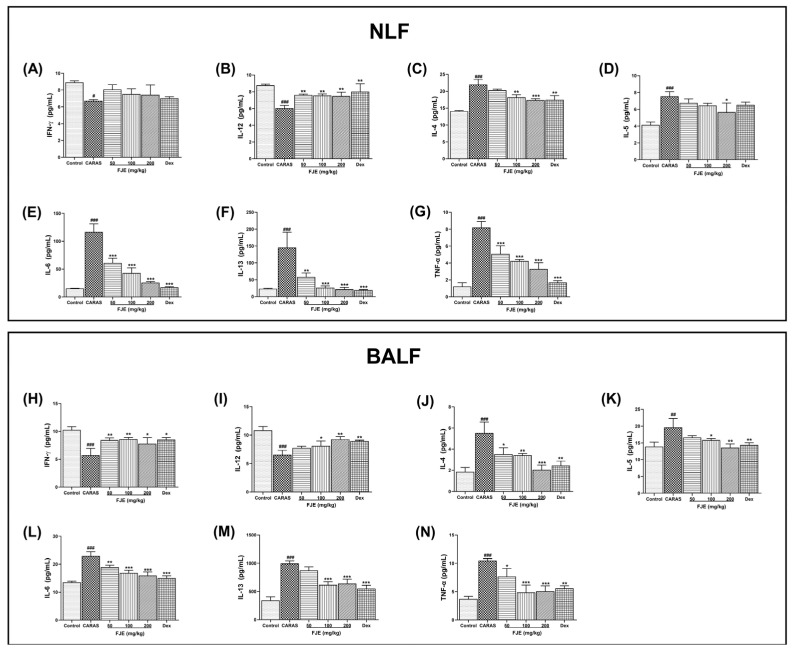
Influence of FJE on Th1- and Th2-associated cytokines in NLF and BALF. (**A**,**B**) The concentrations of Th1-linked cytokines IFN-γ and IL-12 in NLF. (**C**–**G**) Concentrations of Th2-linked cytokines IL-4, IL-5, IL-6, IL-13, and TNF-α in NLF. (**H**,**I**) Concentrations of Th1-linked cytokines IFN-γ and IL-12 in BALF. (**J**–**N**) Concentrations of Th2-linked cytokines IL-4, IL-5, IL-6, IL-13, and TNF-α in BALF. All levels were measured using ELISA kits. Mice were administered saline, Dex (1.5 mg/kg), or FJE (50, 100, or 200 mg/kg) once a day for 16 days. Data represent mean ± SEM (*n* = 6/group). # *p* < 0.05, ## *p* < 0.01, ### *p* < 0.001 compared with control, * *p* < 0.05, ** *p* < 0.01, *** *p* < 0.001 vs. CARAS. Dex, Dexamethasone.

**Figure 6 ijms-24-12514-f006:**
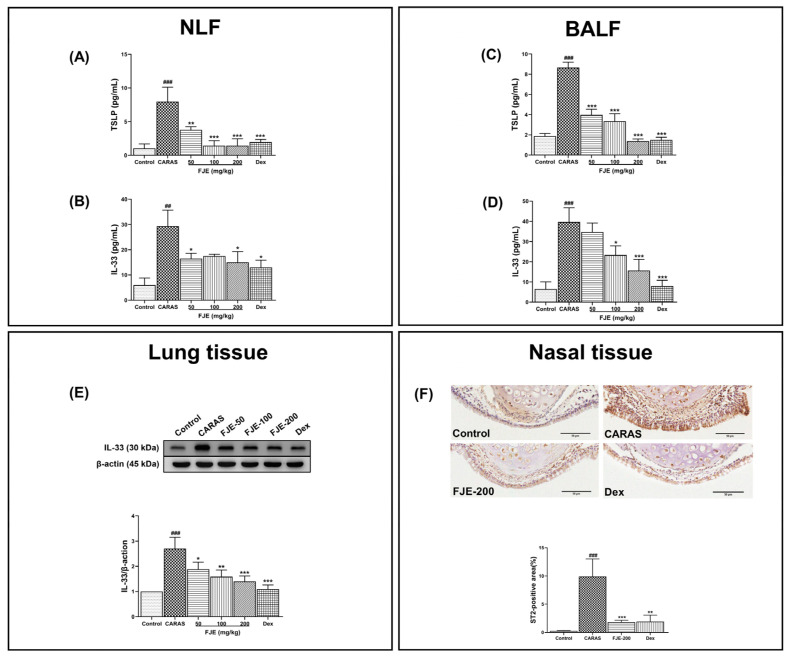
The expression of epithelium-derived cytokines in NLF, BALF, lung, and nasal tissues from the OVA-induced CARAS mouse model. Influence of FJE on (**A**) TSLP and (**B**) IL-33 levels in NLF and (**C**) TSLP and (**D**) IL-33 levels in BALF. (**E**) Immunoblot showing expression of IL-33 in lung tissue and quantification of IL-33. (**F**) Representative IHC images showing ST2 expression in the epithelium layer (dark brown color). Eosinophils are stained dark brown in the cytoplasm. Mice were administered saline, Dex (1.5 mg/kg), or FJE (50, 100, or 200 mg/kg) once a day for 16 days. Data represent mean ± SEM (*n* = 6/group). ## *p* < 0.01, ### *p* < 0.001 compared with control, * *p* < 0.05, ** *p* < 0.01, *** *p* < 0.001 vs. CARAS. Dex, Dexamethasone.

**Figure 7 ijms-24-12514-f007:**
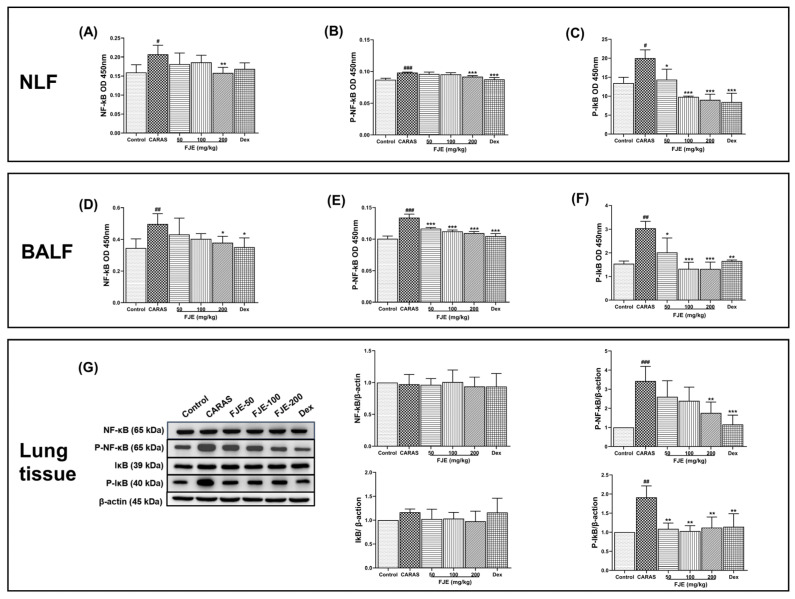
Effect of FJE on the NF-κB signaling pathway in NLF, BALF, and lung tissue. (**A**–**C**) Optical density of levels of NF-κB, P-NF-κB, and P-IκB in NLF. (**D**–**F**) Optical density of levels of NF-κB, P-NF-κB, and P-IκB in BALF. (**G**) Immunoblotting of NF-κB, P-NF-κB, IκB, and P-IκB in lung tissue and quantification of protein levels. Mice were administered saline, Dex (1.5 mg/kg), or FJE (50, 100, or 200 mg/kg) once a day for 16 days. Data represent mean ± SEM (*n* = 6/group). * *p* vs. CARAS and # *p* compared with control. # *p* < 0.05, ## *p* < 0.01, ### *p* < 0.001 compared with control, * *p* < 0.05, ** *p* < 0.01, *** *p* < 0.001 vs. CARAS. Dex, Dexamethasone.

**Figure 8 ijms-24-12514-f008:**
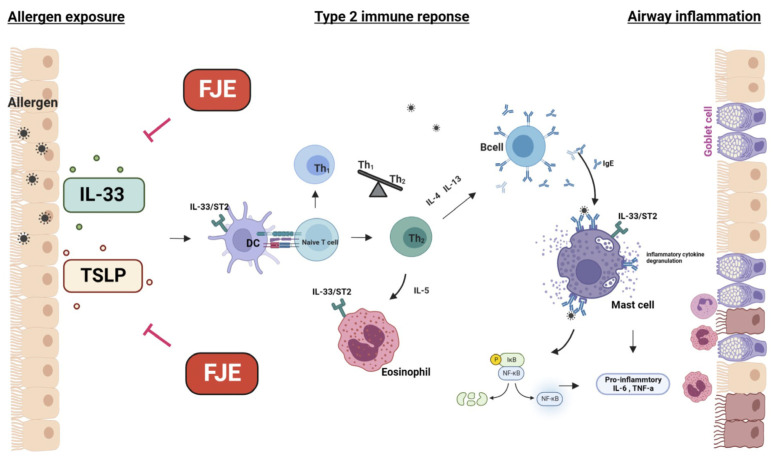
The molecular mechanism underlying the beneficial effects of FJE in the OVA-induced CARAS mouse model. Exposure to allergens induces the release of IL-33 and TSLP from epithelial cells and promotes the differentiation of Th0 cells to Th2. This stimulates B cells to produce IgE, which binds and activates mast cells. Activated mast cells trigger the release of pro-inflammatory cytokines and chemokines. Together with Th2 cell–released cytokines, leukocytes are recruited and activated, leading to allergic inflammatory symptoms. Moreover, the NF-κB pathway is activated to amplify the inflammatory condition. FJE exhibited anti-allergic inflammatory effects via the IL-33/TSLP/NF-κB signaling pathway.

**Figure 9 ijms-24-12514-f009:**
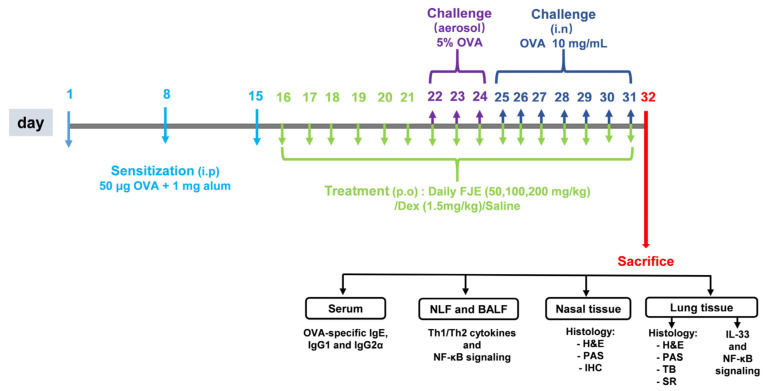
Schematic representation of the experimental design. To generate the OVA-induced CARAS mouse model, mice were sensitized intraperitoneally with OVA on days 1, 8, and 15. FJE/Dex were administered orally 1 h before the intranasal challenge from day 16 to day 31. Mice were administered saline, Dex (1.5 mg/kg), or FJE (50, 100, or 200 mg/kg) once a day for 16 days. Control mice were sensitized with saline.

## Data Availability

The datasets generated during and/or analyzed during the current study are available from the corresponding author on reasonable request.

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
