# Peer review of "Fallopia japonica Root Extract Ameliorates Ovalbumin-Induced Airway Inflammation in a CARAS Mouse Model by Modulating the IL-33/TSLP/NF-κB Signaling Pathway"

_ijms, 2023, doi:10.3390/ijms241512514_

Round 1
Reviewer 1 Report
Comments and Suggestions for Authors
The roots contain polygonin, an anthraquinone derivative, which hydrolyzes to emodin, emodin methyl ether, etc. . These ingredients act as a laxative effect that causes mild diarrhea, a menstrual effect that regulates menstrual irregularities, and a diuretic effect that improves urine flow. It is effective for laxative, diuretic, regular constipation, cystitis, bladder stones, irregular menstruation, and postpartum lochia, and is safe for the elderly and women. used as an oriental medicine. The article presents another novel power of the herbal medicine.
The abstract: “At the molecular level, FJE ameliorated ovalbumin-induced airway inflammation in CARAS model mice by modulating the IL-33/TSLP/NFĸB signaling pathway.”
This sentence should be presented more specifically and in detail.
FJE first appeared in L70 should be typed out.
In Figure 3 C and F, please describe arrows.
In Figure 7F, the microphotographs are out of focus. The positivity in the epithelial ciliated layer might be false positive. Isotype negative controls should be presented.
L286: “Here, Th2 cytokines like IL-4, IL-5, and IL-13 were significantly reduced upon FJE treatment, suggesting improved Treg FOXP3.”
I agree with the comments. It is better to show the evidence that FOXP3 cells are involved in this animal model.
Author Response
We would like to express our sincere thanks to the reviewers who identified areas of our manuscript that needed corrections/formations/ modifications. We have responded to the reviewer’s comments point by point.

Reviewer 2 Report
Comments and Suggestions for Authors
Thank you for the opportunity to familiarize yourself with the work
Please provide the qualitative and nutrient composition of the substance. biologically active in the studied Fallopia japonica
Comments on the Quality of English LanguageThere are many spelling and language mistakes and the manuscript needs to be corrected by a native English speaker.
Author Response
We would like to express our sincere thanks to the reviewers who identified areas of our manuscript that needed corrections/formations/ modifications. We have responded to the reviewer’s comments point by point
